# Interpretation of Near-Infrared Spectroscopy (NIRS) Signals in Skeletal Muscle

**DOI:** 10.3390/jfmk4020028

**Published:** 2019-05-26

**Authors:** Adeola A. Sanni, Kevin K. McCully

**Affiliations:** Department of Kinesiology, University of Georgia, Athens, GA 30602, USA

**Keywords:** near-infrared spectroscopy (NIRS), oxygen consumption, hemoglobin, myoglobin, skeletal muscle

## Abstract

Near-infrared spectroscopy (NIRS) uses the relative absorption of light at 850 and 760 nm to determine skeletal muscle oxygen saturation. Previous studies have used the ratio of both signals to report muscle oxygen saturation. Purpose: The purpose of this pilot study is to assess the different approaches used to represent muscle oxygen saturation and to evaluate the pulsations of oxygenated hemoglobin/myoglobin (O_2_heme) and deoxygenated hemoglobin/myoglobin (Heme) signals. Method: Twelve participants, aged 20–29 years, were tested on the forearm flexor muscles using continuous-wave NIRS at rest. Measurements were taken during 2–3 min rest, physiological calibration (5 min ischemia), and reperfusion. Ten participants were included in the study analysis. Results: There was a significant difference in pulse size between O_2_heme and Heme signals at the three locations (*p* < 0.05). Resting oxygen saturation was 58.8% + 9.2%, 69.6% + 3.9%, and 89.2% + 6.9% when calibrated using O_2_heme, the tissue oxygenation/saturation index (TSI), and Heme, respectively. Conclusion: The difference in magnitude of O_2_heme and Heme pulses with each heartbeat might suggest different anatomical locations of these signals, for which calibrating with just one of the signals instead of the ratio of both is proposed. Calculations of physiological calibration must account for increased blood volume in the tissue because of the changes in blood volume, which appear to be primarily from the O_2_heme signal. Resting oxygen levels calibrated with Heme agree with theoretical oxygen saturation.

## 1. Introduction

Skeletal muscle contractions play an essential role in human locomotion by generating the forces required for movement [1]. A key to muscle contractions is the production of chemical energy (ATP) primarily from oxidative metabolism in muscle mitochondria [2,3]. Changes in oxygen levels in skeletal muscle can provide insights into the function of the skeletal muscle in both healthy and diseased states. Near-infrared spectroscopy (NIRS) has been increasingly used to study oxygen levels in skeletal muscle [4,5,6,7]. A search of PubMed in January 2019 for the terms “NIRS” and “muscle” showed 679 publications, 126 of which are from the last two years. NIRS measurements of muscle oxygen level, blood flow, and metabolism are noninvasive and have shown good agreement with other measurement modalities [8]. Continuous-wavelength NIRS devices use physiological calibration (vascular occlusion) to 0%; this has been shown to allow accurate quantitative measurements in the skeletal muscle [4].

Several methodological questions about NIRS measurements of skeletal muscle remain [7,9], such as which of the NIRS signals is the most appropriate to report, and if physiological calibration (5 min ischemia) reaches 0% oxygen saturation. NIRS measurements typically record absorbance of light at several wavelengths, where changes in absorbance in the region near 850 nm are ascribed to oxygenated hemoglobin/myoglobin (O_2_heme), and absorbance in the region near 760 nm is attributed to deoxygenated hemoglobin/myoglobin (Heme). Often, a ratio of absorbance at 850/760 nm or 850/(850 + 760) nm is used to describe oxygen saturation [10]. A popular ratio is the tissue oxygenation/saturation index (TSI), which is the ratio of absorbance at 850/(850 + 760) nm × 100 to produce percentage values [11]. However, some investigators, such as Grassi et al. [7], have proposed using only the Heme signal because it seems to better reflect oxygen extraction in a variety of experimental approaches. Thus, there are still questions regarding the most appropriate NIRS signal to report. There are also questions as to whether the NIRS pulsation signal in skeletal muscle reflects contributions from hemoglobin or myoglobin [7,12]. Some investigators have suggested that myoglobin represents 10% of the NIRS signal [13,14], while others have reported it reflects 80% of the signal [15]. Davis et al. [9] reported approximately 32% hemoglobin in the NIRS signal in human skeletal muscle and that this value depends on anatomical and experimental positions. 

The purpose of this study was to evaluate NIRS signals from 760 and 850 nm in response to ischemia and reperfusion in the forearms of healthy young adults. We hypothesized that: (1) blood volume change alters the physiological calibration using the O_2_heme signal, producing a lower-than-expected oxygen saturation value; (2) the physiological calibration produces O_2_heme values that can be considered zero oxygen saturation; and (3) pulsations in the NIRS signals due to heart rate reflect oxygen saturation specifically from hemoglobin. 

## 2. Materials and Methods 

### 2.1. Participants

This project involved a convenience sample of University of Georgia students who volunteered as study participants. Twelve subjects (five males, seven females) aged 20–29 years participated in the study. Ten subjects were included in the final analysis, as two participants’ data were excluded because the electrical stimulation current did not activate the muscle and increase the metabolic rate as required (at least a fourfold increase). The study was conducted with the approval of the Institutional Review Board at the University of Georgia (Athens, GA, USA). All subjects gave their informed consent for inclusion before participation. The study was conducted in accordance with the Declaration of Helsinki, and the protocol was approved by the Ethics Committee of the University of Georgia (Protocol #STUDY00004412) on 11 February 2017.

### 2.2. Experimental Design

This pilot study used a single group design. Comparisons were made between three different experimental conditions (rest, ischemia, and reperfusion) performed during one testing session. The protocol consisted of measuring muscle oxygen consumption at rest, during 5 min of ischemia, and evaluating NIRS signals during reactive hyperemia after the release of ischemia. 

### 2.3. Experimental Procedures 

The measurement was done on the forearm flexor muscles with the subjects in a supine position. A continuous-wave NIRS device (Portamon, Artinis Medical Systems, Einsteinweg, Netherlands) was placed on the proximal/medial portion of the forearm [16], which was put in place with a nonelastic wrap. Adipose tissue thickness was measured using ultrasound (LOGIQ, GE Healthcare) as previously described [17]. NIRS measurements were digitally recorded throughout the protocol at an acquisition frequency of 10 Hz. 

Neuromuscular electrical stimulation was used to increase muscle metabolic rate. Electrodes (2 × 2 cm) were placed proximally and distally to the NIRS device and connected to a commercial electrical stimulation device (RICH-MAR, theratouch 4.7, Version 15, Chattanooga, TN, USA). The muscle was stimulated at 6 Hz. Biphasic square wave pulses (200 µs with a 50 µs interpulse delay) with submaximal current levels (25–40 mA) tolerable for each subject were used to activate the muscle to provide an increase in the metabolic rate [16]. 

Muscle ischemia was produced with complete vascular occlusions with a blood pressure cuff, (Hokanson, Bellevue, WA, USA) placed about 2 cm above the elbow. The pressure of the cuff was set to 220–260 mmHg using a rapid cuff inflation system (Hokanson, E20 Rapid Cuff Inflator and a 30 gal capacity commercial air compressor).

### 2.4. Testing Protocol

After a rest period of 2–3 min, the blood pressure cuff was inflated for 30 s to measure the rate of muscle oxygen consumption, and a 2 min baseline measure was recorded. Electrical stimulation was introduced for 30 s to activate the muscle and check the rate of increase in muscle oxygen consumption; this helped identify the current level that was sufficient to activate each participant’s muscle. Physiological calibration was performed (5 min ischemia). Cuff inflation for 5 min was preceded by electrical stimulation using the previously identified current to increase metabolic rate and reduce the time needed to reach full ischemia. Electrical stimulation was also done for 30 s to the end of the ischemia to check if oxygen consumption reached 0%. The cuff was deflated and measurements were taken until the signal reached peak reactive hyperemia. Figure 1 shows the example graph of the protocol.

### 2.5. Data Analysis 

Oxygen consumption and oxygen delivery were measured using O_2_heme and Heme, respectively. The raw data collected from the NIRS device were exported and analyzed on Microsoft Excel. A graph of the NIRS signal (optical density, OD) was plotted against time (calculated using the frequency of data collection). The measurement was recorded from the third channel of the NIRS signal; this was done to avoid the influence of adipose tissue thickness at the shallow channel [18]. OD was recorded at different points of oxygen consumption. The percent pulse size from each NIRS signal was measured and compared at rest immediately after ischemia (early reperfusion) and during peak reactive hyperemia. Pulse size was calculated by finding an average of three consecutive pulsations (wave heights) and subtracting the average of the two minimum pulse signals (troughs) from the maximum pulse signal (crest), multiplied by 100; values are in OD. All measurements were calibrated (physiological calibration) with the delta range of reactive hyperemia after 5 min of ischemia. Pulse size was also calculated as 32% of hemoglobin relative to total Heme (hemoglobin plus myoglobin), as recommended by Davis and Barstow [9].

### 2.6. Statistical Analysis 

Data were analyzed using IBM SPSS Statistics software v24 (IBM®, Armonk, NY, USA). One-way ANOVA was used to identify the difference among values of resting oxygen saturation calculated using O_2_heme, Heme, and TSI signals, and a pairwise Bonferroni post hoc comparison was made to evaluate the individual paired difference. A paired sample *t*-test was used to test the difference in oxygen saturation before and after electrical stimulation during the physiological calibration (5 min cuff). A 3 × 2 factor ANOVA was used to identify the difference between O_2_heme and Heme pulse size at rest immediately after ischemia and during peak reactive hyperemia. The significance level was set at ≤0.05 (two-tailed) for all comparisons with post-hoc adjustments used as needed for multiple comparisons.

## 3. Results

The characteristics of the participants in this study are shown in Table 1. An example protocol used in this study is shown in Figure 1.

### 3.1. Oxygen Saturation

Figure 2A shows reactive hyperemia using the Heme and O_2_heme signals, including the difference in both signals (blood volume). There was a significant increase in blood volume during reperfusion, which was about 16% ± 6% in the O_2_heme signal. Resting oxygen saturation was calculated using three different methods (Figure 2B). There was a significant difference among the three values (F(2,27) = 48.2, *p* < 0.001). Pairwise comparisons showed significant individual paired difference between the O_2_heme, TSI, and Heme signals (*p* < 0.05 for all comparisons). 

### 3.2. Physiological Calibration

With ischemia, the O_2_heme signal decreased and the Heme signal increased, showing an increase in deoxygenated blood and a decrease in oxygenated blood due to muscle oxygen consumption. There were no significant changes in either signal with electrical stimulation after 4–5 min of ischemia (O_2_heme signal, *p* = 0.148; Heme signal, *p* = 0.598). Figure 3 shows the mean difference and confidence interval of oxygen level before and after stimulation during the 5 min ischemia. TSI reached approximately the minimum (45% ± 11%) and maximum (76% ± 5%), respectively.

### 3.3. NIRS Signal Pulsation

Representative images of heart-rate-induced pulse sizes from both O_2_heme and Heme signals are shown in Figure 4. Pulse sizes were calibrated to the ischemic/hyperemia range and the assumed percent of hemoglobin in the NIRS signals [9]. There was a significant interaction between the three locations and the two NIRS signals ($F(2,36=7.76,\;\eta^{2}=0.30,\;p<0.01$). There was a significant difference among the pulse sizes at the three locations (rest, cuff end, and hyperemia) $\left(F\left(2,36\right)=75.9,\;\eta^{2}=0.81,\;p<0.001\right)$. Bonferroni multiple correction showed a significant difference between each pair of locations (*p* < 0.001). There was a significant main effect difference between O_2_heme and Heme pulse size ($F(1,18=60.9,\;\eta^{2}=0.77,\;p<0.001$). Pulse sizes were larger for the O_2_heme signal compared with the Heme signal and were largest later during reactive hyperemia compared with during early reperfusion and when at rest (Figure 5A). The ratios of pulse sizes between O_2_heme and Heme were not similar to the oxygen saturation values for the NIRS signals at the three locations (Figure 5B). 

## 4. Discussion

This study evaluated the use of NIRS signals to determine oxygen levels in skeletal muscle. NIRS signals at 850 and 760 nm light were presented as O_2_heme and Heme to reflect the contribution of myoglobin and hemoglobin. Previous studies have presented NIRS signals as O_2_Hb and HHb, or as Hb/Mb [4,5,6,7]. This was done based on observations that the NIRS signals come from both hemoglobin and myoglobin and the need to provide more precise terminology for NIRS signals [19].

### 4.1. Appropriate NIRS Signal for Oxygen Saturation in the Muscle

This study found that physiological calibration includes a transient increase in blood volume, which appears to be entirely from the O_2_heme signal. Calculating a resting oxygen saturation value using O_2_heme or a ratio of both O_2_heme and Heme (such in TSI) resulted in oxygen saturation values that were lower than values obtained if only the Heme signal was used. The resting oxygen saturation values (~59%) found with the O_2_heme difference signal was similar to that seen in previous studies (<70%) [20,21]. However, using only the Heme signal for physiological calibration resulted in resting oxygen saturation values of 91%. This value is consistent with the expected value of Heme oxygen saturation (~88%) based on the assumptions that, at rest, myoglobin oxygen saturation is 100% [22], the myoglobin contribution to the total Heme in muscle is 70%, and there are hemoglobin saturation values of 70% (between 98% in the artery and 40% in the veins) and a hemoglobin contribution to the total Heme of 30%. For the resting oxygen saturation of the total Heme to be near 60%, either there must be significant myoglobin desaturation at rest or hemoglobin oxygen saturation needs to be less than zero, neither of which is supported in the literature. The use of the deoxygenated signal (Heme) to reflect changes in oxygen levels in muscle has been suggested previously [7,23,24]. Interestingly, TSI is a commonly used approach to present oxygen saturation values, and in our study, resting TSI was 70%, which is consistent with previous studies. TSI also does not agree with the calculated Heme oxygen saturations (above). TSI is determined from the ratio of light absorbance at the two wavelengths and never approaches either zero or 100% oxygen levels during a physiological calibration. Therefore, we propose that oxygen saturation should be calibrated with the (physiological calibration) ischemia protocol [25] using the Heme signal.

### 4.2. Physiological Calibration Using 5 Min Ischemia with Prior Exercise

This study found that 5 min of ischemia with a prior 30 s of electrical stimulation resulted in a minimal value of Heme oxygen levels. This was shown by the lack of change in either the O_2_heme or the Heme signals after an additional electrical stimulation once the signals had plateaued. Previous studies have not always found that 5 min of ischemia produced minimal oxygen levels or maximal reactive hyperemia. Five minutes of ischemia has produced an 80–90% change in oxygen saturation levels [26] and about 80–90% of the maximal hyperemic blood flow response measured with ultrasound [27]. However, the use of prior exercise to increase metabolic rate (in this study, it produced an increase in metabolic rate approximately fivefold above resting metabolic rate) did appear to result in complete desaturation of the muscle. The prior use of exercise or electrical stimulation to increase metabolic rate has been proposed to assure a maximal change in oxygen levels and blood flow [28].

### 4.3. Interpretation of the Pulsatile O_2_heme and Heme Signals from Muscle 

We could not accept the hypothesis that pulsations in the NIRS signals reflect the oxygen saturation from the entire hemoglobin signal. Because the pulsations are due to changes in the hemoglobin signal and not the myoglobin signal, the hypothesis was that the ratio of the signal size of the O_2_heme and Heme pulsations would reflect hemoglobin oxygen saturation. However, the ratio of the signal size of the O_2_heme and Heme pulsations were significantly different from the hemoglobin oxygen saturation values that would be expected during immediate reperfusion at low oxygen levels as well as during reactive hyperemia once oxygen levels were maximal. Pulsations in blood due to beating of the heart are thought to disappear in capillaries and venules [29]; however, more recent studies have found evidence of pulsations in skeletal muscle capillaries [30]. An alternative hypothesis is that the O_2_heme and Heme pulsations represent oxygen saturation levels on the arterial side of the microvascular system. The size of the pulses relative to the estimated total hemoglobin concentrations is small, although this increases during reactive hyperemia when vascular tone is decreased. The presence of pulsations in the Heme signal during peak reactive hyperemia suggests that the 100% value for total Heme from the physiological calibration is actually less than 100%. This is to be expected, as the muscle is still consuming oxygen even during the peak reactive hyperemia time period. However, the size of the Heme pulsations is small enough to allow the assumption of 100% to be close to accurate, and it would be difficult to accurately determine how much different the actual value would be from 100%. Thus, the pulsations in the O_2_heme and Heme signals from NIRS most likely come from precapillary arterioles [31], although changes in vascular tone might alter the microvascular area that contributes to the signal [4].

### 4.4. Limitations

This study was performed using one type of NIRS device: the continuous-wavelength “Portamon” from Artinis Ltd. Most continuous-wavelength devices use similar wavelengths and calculations to determine O_2_heme and Heme signals. While the results might be expected to be similar for other continuous-wavelength devices, other devices use phase modulation of photon counting to determine both absorption and scattering of light, allowing more accurate calculations of oxygen levels. How these devices determine O_2_heme and Heme might be different enough to produce different relationships between the variables. We used an assumption of the relative contribution of myoglobin and hemoglobin in human muscle based on Davis and Barstow [9]. This assumption was used as a general approximation, as hemoglobin levels can vary between people and experimental conditions [9]. We also tested our subjects in a supine position and did not alter body position during the experiment. Studies where the body is in the standing position, there are changes in body position, or the muscle of interest is at a different height than the heart might have different changes in blood volume from our finding. Under those circumstances, the changes in blood volume might not reflect purely O_2_heme as they did in our study. This study included a relatively small sample size of 10 healthy young male and female participants. Future studies could include more heterogeneous populations, including those with clinical conditions, so the findings in the study can be generalized to a larger population. Also, to the best of our knowledge, no study has shown any difference in muscle oxygen between sex, but future studies can consider using sex as a biological variable during NIRS analysis.

## 5. Conclusions

Muscle oxygen levels provide an important window into the function of skeletal muscle. Near-infrared spectroscopy has been increasingly used to evaluate skeletal muscle oxygen levels in both health and disease. This study found that NIRS measurements using a physiological calibration consisting of ischemia with prior exercise can determine a range of oxygen levels in muscle that goes from 0–100%. Calculations of a physiological calibration must account for increased blood volume in the tissue due to reactive hyperemia. Because of the changes in blood volume, which appear to be primarily from the O_2_heme signal, the Heme signal is perhaps a better signal with which to perform the calibration. Therefore, identifying the most appropriate NIRS signal to report during measurement of skeletal muscle oxygenation will be beneficial for researchers using NIRS and also clinical practitioners who use the device to measure skeletal muscle oxygenation. Finally, NIRS-based signals have pulsations in signal intensity related to heart rate. These pulsations most likely reflect hemoglobin in the atrial side of the microvascular system and not the entire hemoglobin signal. The conclusions of this pilot study should help improve the interpretation and usefulness of NIRS measurements of skeletal muscle.

## Figures and Tables

**Figure 1 jfmk-04-00028-f001:**
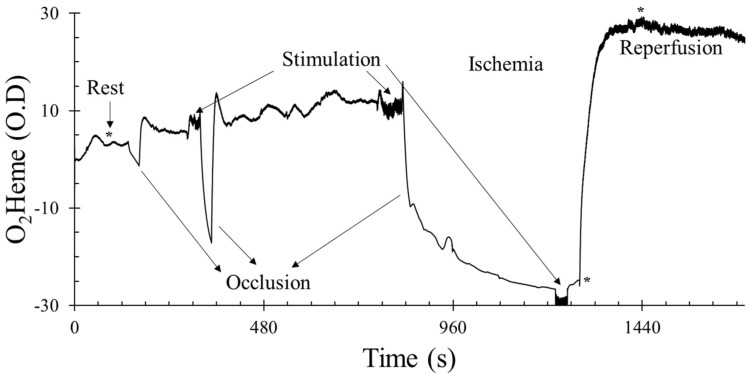
An example of the testing protocol using the oxygenated hemoglobin/myoglobin (O_2_heme) signal. The *y*-axis scale is in optical density units. * shows the approximate time points when pulse size was calculated at rest, immediately after ischemia, and during reactive hyperemia. O_2_heme and deoxygenated hemoglobin/myoglobin (Heme) physiological calibration reached 0% during ischemia and 100% at peak hyperemia.

**Figure 2 jfmk-04-00028-f002:**
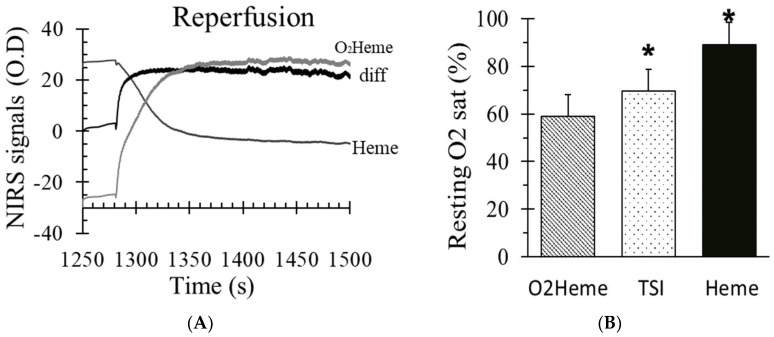
(**A**) The O_2_heme and Heme signals during reactive hyperemia and the difference in both signals which indicated the influence of blood volume change. (**B**) The percent of oxygen saturation at rest from three different methods of calculating oxygen saturation (O_2_heme, tissue oxygenation/saturation index (TSI), and Heme). * indicates a significant difference in the method of calculating oxygen saturation (*p* < 0.01 for the three comparisons). Values are means ± SD.

**Figure 3 jfmk-04-00028-f003:**
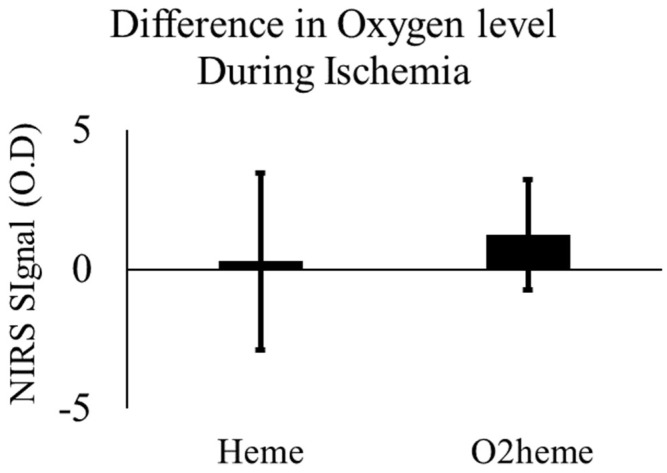
The change in O_2_heme and Heme signals before and after stimulation during the 5 min of ischemia with prior stimulation. There was no significant difference, *p* > 0.05. Values are means with the 95% confidence intervals.

**Figure 4 jfmk-04-00028-f004:**
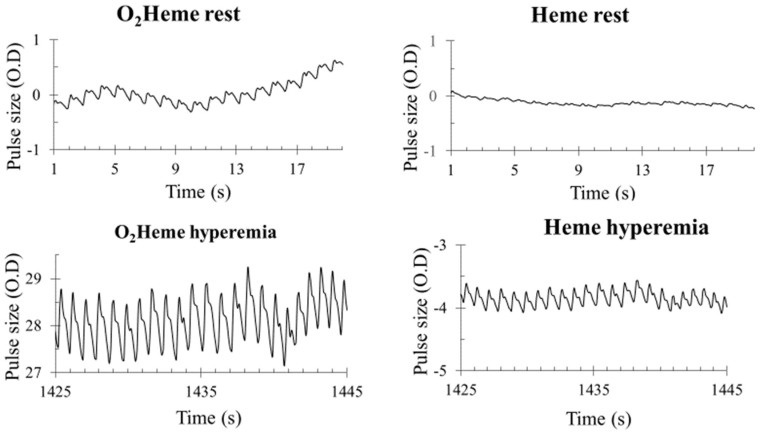
Examples of blood flow pulsations of O_2_heme and Heme signals at rest and during peak reactive hyperemia. The *y*-axis scale is in optical density units.

**Figure 5 jfmk-04-00028-f005:**
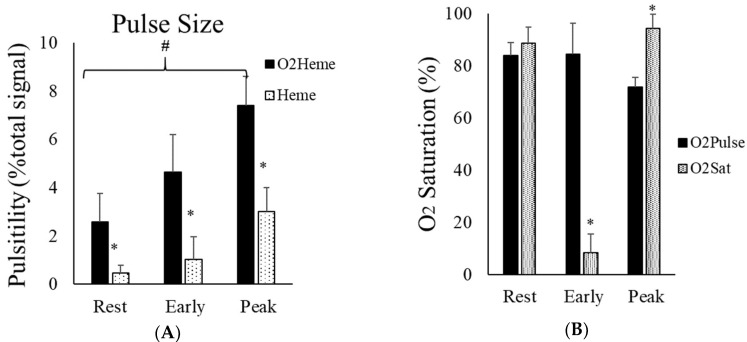
(**A**) The average pulse size signals of O_2_heme and Heme at rest, immediately after ischemia, and during reactive hyperemia. The *y*-axis scales are in percentage of the calculated physiological calibration. # indicates a significant difference in the pulse size at each location. * indicates a significant difference between the O_2_heme and Heme signals. *p* < 0.001 for all comparisons. (**B**) The ratio of the pulse size at the three different locations and the oxygen saturation calibrated to each signal at the same location. * shows the difference between ratios of pulse and oxygen saturation at each location. Values are means ± SD.

**Table 1 jfmk-04-00028-t001:** Characteristics of study participants.

	Male	Female
*N*	5	5
Age (years)	23.6 (4.3)	20.2 (0.25)
Height (cm)	170 (0.2)	1.66 (0.04)
Weight (kg)	73.4 (12.3)	64.22 (10.5)
Body Mass Index (kg/m^2^)	25.2 (3.4)	23.51 (4.9)
Adipose Tissue Thickness (cm)	0.34 (0.09)	0.40 (0.22)

Values are means (standard deviations).

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
