# Peer review of "Interpretation of Near-Infrared Spectroscopy (NIRS) Signals in Skeletal Muscle"

_jfmk, 2019, doi:10.3390/jfmk4020028_

Round 1
Reviewer 1 Report
Great manuscript: clear, easy to follow, well constructed, good figures. Congratulations.
Author Response
Thank you for taking out time to read and review our paper, we read through the paper to correct for minor English spell check. We hope everything looks good now.
Reviewer 2 Report
Manuscript titled “interpretation of near-infrared spectroscopy (nirs) signals in Skeletal muscle” deals an important issue of medical muscle biology. The purpose of this study was to evaluate the NIRS signals from 760nm and 850nm in 61 response to ischemia and reperfusion in the forearms of healthy young adults.
The work is good and interesting but in the present form is poor details and lack of important information. The experiments are adequate and the results are well presented. Moreover, the state of the art/background is very poor, also there are some minor and major concerns that need to be addressed before recommending publication.
Please rewrite the abstract in a clearer form, adding the aim of this study.
The authors could be adding a graphical abstract to explain in a better fast way the aim and the design of this study and to help better readers understand.
Please reformulate and rewrite the introduction section, it misses important and fundamental details about muscle biology and functions. It must be improved and updated. In the present form is too poor for this huge scientific field of research. The authors should go deeper into the different and relevant aspect of the muscle. I recommend checking the following recent and interesting papers and refer to them in relation to the study topic:
Somitogenesis: From somite to skeletal muscle. Acta Histochem. 2015 May-Jun;117(4-5):313-28.
Effects of dietary extra-virgin olive oil on oxidative stress resulting from exhaustive exercise in rat skeletal muscle: a morphological study. Acta Histochem. 2014 Jan;116(1):61-9.
Morphological and Functional Aspects of Human Skeletal Muscle. J. Funct. Morphol. Kinesiol. 2016, 1, 289-302.
Methods
How the number of participants (n=12) was determined? The power analysis sample size was performed?
The number of participants is insufficient to conclude, as Authors had in their study twelve subjects (5 males, 7 females) of various age 20-29yrs. So, based on such number of participants, they cannot have reliable observations, due to natural variability (gender and age).
Please add more information about ethical approval.
In conclusion please specify the clinical relevance of your work and some important suggestions for the scientific community.
Author Response
Manuscript titled “interpretation of near-infrared spectroscopy (NIRS) signals in skeletal muscle” deals an important issue of medical muscle biology. The purpose of this study was to evaluate the NIRS signals from 760nm and 850nm in 61 response to ischemia and reperfusion in the forearms of healthy young adults.
The work is good and interesting but in the present form is poor details and lack of important information. The experiments are adequate and the results are well presented. Moreover, the state of the art/background is very poor, also there are some minor and major concerns that need to be addressed before recommending publication.
We appreciate the time and effort of the reviewer in evaluating our manuscript. We feel the suggestions of the reviewer will improve the paper, and we hope the revision adequately addresses the reviewers concerns.
Please rewrite the abstract in a clearer form, adding the aim of this study.
We have reviewed and revised the introduction. We hope that the revision is clearer in its background and purpose.
The authors could be adding a graphical abstract to explain in a better fast way the aim and the design of this study and to help better readers understand.
We are unfamiliar with graphical abstracts, but we have attempted to make one. We are agree that a graphical abstract could be useful.
Please reformulate and rewrite the introduction section, it misses important and fundamental details about muscle biology and functions. It must be improved and updated. In the present form is too poor for this huge scientific field of research. The authors should go deeper into the different and relevant aspect of the muscle. I recommend checking the following recent and interesting papers and refer to them in relation to the study topic:
We have cited one of the papers listed by the reviewer, and hope that will help the reader to better understand our paper.
Morphological and Functional Aspects of Human Skeletal Muscle. J. Funct. Morphol. Kinesiol. 2016, 1, 289-302.
Methods
How the number of participants (n=12) was determined. The power analysis sample size was performed?
The number of participants is insufficient to conclude, as Authors had in their study twelve subjects (5 males, 7 females) of various age 20-29yrs. So, based on such number of participants, they cannot have reliable observations, due to natural variability (gender and age). ?
We agree with the reviewer that adequate sample sizes and uniform subject characteristics are important. In Student T-Tests, we typically find that sample sizes between 8-10 provide sufficient power to make useful comparisons. Based on previous studies, we have not found significant differences between young men and young women in this age group. We also saw (by observation) not numerical differences in our results between the male and female subjects. The age range of 20-29 is typically thought to not result in significant age related changes. Because our subjects were all healthy and moderately active, we feel we had a relatively uniform sample.
Please add more information about ethical approval.
We have included the standard two statements (the study was approved by the University of Georgia and all subjects provided informed consent).
In conclusion please specify the clinical relevance of your work and some important suggestions for the scientific community.
We agree that clearly stating clinical relevance is important, and we have revised our conclusions to address this point.

Round 2
Reviewer 2 Report
The authors reply in adequate way to my comments. Only one minor revision: Please replaced twelve subjects to ten in section participants.
Author Response
Thank you for taking out time to review our paper.
The minor correction has been included in the paper methods section
Twelve subjects participated in the study but ten were included in the analysis because for two of the participant, electrical stimulation used was not enough to increase metabolic rate as required (line 18 and line 76-78)